# How Instrument Transformers Influence Power Quality Measurements: A Proposal of Accuracy Verification Tests

**DOI:** 10.3390/s22155847

**Published:** 2022-08-04

**Authors:** Gabriella Crotti, Yeying Chen, Huseyin Çayci, Giovanni D’Avanzo, Carmine Landi, Palma Sara Letizia, Mario Luiso, Enrico Mohns, Fabio Muñoz, Renata Styblikova, Helko van den Brom

**Affiliations:** 1Istituto Nazionale di Ricerca Metrologica (INRIM), Str. delle Cacce, 91, 10135 Torino, Italy; 2Physikalisch-Technische Bundesanstalt (PTB), Parkplatz, Bundesallee 100, 38116 Braunschweig, Germany; 3Turkiye Bilimsel ve Teknolojik Arastirma Kurumu (TUBITAK), Tunus Caddesi No:80, Kavaklıdere, Ankara 06680, Turkey; 4Dipartimento di Ingegneria, Università degli Studi della Campania “Luigi Vanvitelli”, Via Roma 29, 81031 Aversa, Italy; 5Ricerca sul Sistema Energetico-RSE S.p.A, Via Rubattino, 54, 20134 Milano, Italy; 6VSL B.V. (VSL), Thijsseweg 11, 2629 JA Delft, The Netherlands; 7Cesky Metrologicky Institut (CMI), Okružní 31/772, 638 00 Brno, Czech Republic

**Keywords:** power system measurements, instrument transformers, power quality, accuracy

## Abstract

The integration of renewable energy sources on a large scale in the electrical energy distribution systems, as well as the widespread of non-linear loads, has led to a significant increase in power quality (PQ) disturbances. For this reason, PQ monitoring is also becoming a key task in medium voltage (MV) grids. The measurement of PQ at MV levels can only be performed using instrument transformers (ITs) to scale down the level of voltage and current to levels suitable for the input stage of PQ instruments. However, no international standards currently require the verification of the errors introduced by ITs in the measurement of PQ phenomena. Moreover, this issue is only partially addressed in the scientific literature, where papers dealing with specific and limited aspects of the problem can be found. For this reason, this paper aims to comprehensively assess the issue, proposing IT accuracy verification tests for different PQ parameters. First, a set of PQ phenomena relevant for IT testing is chosen, as well as the associated ranges of variation, based on a review of the enforced standards and the scientific literature. For each selected PQ phenomenon, possible performance indices and test waveforms are proposed. Finally, the proposed procedure is validated by applying it to the characterization of two different types of commercial voltage transformers.

## 1. Introduction

Extensive integration of renewable energy sources in Europe’s electrical energy distribution system is essential for the realization of a resource-efficient, green, and competitive low-carbon economy. However, the switching power inverters required to connect renewable energy sources to the supply grid also lead to an increase in grid-injected disturbances [1]. This degradation of power quality (PQ) leads to significant costs to industry and to consumers. Accurate measurement of PQ is thus of critical importance for monitoring the grid. Therefore, PQ measurement methods [2] and testing methods for PQ analyzers to perform these measurements [3] are very well standardized.

Distribution grids typically have rated voltages up to 36 kV, conduct currents up to 2 kA, and show signals containing frequency content up to 9 kHz. PQ monitoring of distribution grids therefore requires connecting PQ analyzers to instrument transformers (Its), namely voltage and current transformers (VTs and CTs), to convert these high voltage and current signals into suitable voltage signals that can be fed to the input stage of PQ analyzers. However, due to their operating principles, Its will not only scale down the voltage and current signals at the power frequency of 50 Hz or 60 Hz, but will also exhibit specific behavior at other frequencies and will show a specific response to PQ phenomena. Consequently, when performing reliable PQ surveys in distribution grids, the behavior of the required Its needs to be accurately established. Nevertheless, so far, standardization of ITs is limited to requirements and methods defined at power frequency only. IEC TC 38 “Instrument Transformers” has recognized this lack of standardization and expressed specific needs for metrology research towards suitable testing methods for Its used for PQ measurements.

The definition of a method for the assessment of Its accuracy in the measurement of PQ is an issue only partially addressed in the scientific literature. Only papers dealing with specific and limited aspects of the problem can be found. For instance, in [4], the impact of VTs in the measurement of amplitude and phase modulation and in the presence of a real voltage dip is estimated. The papers [5,6] investigate the error contributions of VTs and CTs in the measurement of first harmonics. The paper [7] deals with the study of inductive VT performance in the measurement of transient overvoltage. However, a comprehensive analysis of IT performance in the presence of all the PQ phenomena is still lacking, as well as a proposal of a related measurement procedure.

The activity presented here is developed according to the framework of the European Metrology Programme for Innovation and Research (EMPIR) 19NRM05 “IT4PQ” project that aims to develop “Measurement methods and test procedures for assessing the accuracy of Instrument Transformers for Power Quality Measurements” [8]. Specifically, it aims to develop PQ indices to assess the performance of Its, related test procedures, requirements for reference setups for the tests and methods for the evaluation of the uncertainty contributions of Its to PQ measurements.

To this end, suitable performance indices, which quantify the IT errors in the measurement of a single PQ parameter, need to be defined. Furthermore, an overall “aggregated” performance index that can directly qualify the Its performance in PQ measurements would be very helpful for the classification of Its for PQ measurements. The definition of different accuracy classes, and related limits, for this synthetic performance index will be based on deep knowledge and classification of the errors introduced by the Its when measuring the PQ parameters. The introduction of this synthetic index and its associated accuracy classes will allow the extension of the concept of accuracy class, by the introduction of a new, PQ-specific accuracy class index. In fact, currently, the accuracy class is defined in the relevant IEC 60044 and 61869 standards [9,10,11,12,13] as the permissible percentage voltage and current errors only for the rated frequency.

As a starting point, based on a thorough review of the scientific literature and existing standards, the PQ phenomena to be included in this study and their relevant characteristics in terms of voltage or current behavior, in the frequency domain or the time domain, have been defined [14]. 

Starting from the review activity presented in [14], this paper provides possible ranges of variations of the relevant PQ phenomena for IT testing.

Then, we present suitable performance indices as well as test waveforms for the quantitative evaluation of the IT performance when measuring different PQ phenomena including stationary and dynamic as well as transient phenomena, as defined in the IEEE standard 1159 [15]. 

The proposed procedure, in terms of test waveforms and performance indices, is finally validated by applying it to the characterization of two commercial VTs based on different operating principles, an inductive VT and a capacitive low power voltage transformers (LPVT).

The structure of the paper is as follows. Section 2 briefly presents a review of the literature and standards relevant for testing of ITs used for PQ measurements and proposes the range of variation of the PQ phenomena considered. In Section 3, the performance indices for the most relevant PQ parameters for IT testing are proposed. Measurement setups for IT characterization are described in Section 4. In Section 5, the experimental results are presented, demonstrating the usefulness of the proposed performance indices. Finally, in Section 6, the conclusions are drawn.

## 2. Analysis of the Literature and Standards regarding ITs and PQ: Proposed Range of Variation

This section aims to provide a brief overview of the literature and international standards related to ITs and PQ and to identify the PQ phenomena that are relevant for ITs testing. More details on this review can be found in [14].

### 2.1. Literature and Standards Regrading PQ

The study of PQ issues is a matter of great interest in several fields, and, for this reason, the related scientific literature and international standards are very extensive. The international technical standards [2,3,15,16,17] on PQ describe the characteristics of voltage waveforms at a given point of low voltage (LV), medium voltage (MV) and high voltage (HV) public power networks, with the assessing methods for PQ phenomena. The scientific literature [18,19,20,21,22,23,24,25,26,27] is mainly focused on measurement campaigns with a statistical approach, using the standard assessment methods. In the following, a list of the considered PQ disturbances is provided. For each of them, the relevance of testing ITs under the considered events and the possible range of test parameters are discussed.

#### 2.1.1. Frequency Deviation

The frequency deviation is defined by the standard [16] as a variation of the rated frequency value of the supply voltage in a distribution or transmission system of electrical energy. For this type of phenomenon, there is a mismatch between the admissible range of variation of the power frequency and the frequency testing points prescribed by the relevant standards for inductive ITs (power frequency *f*_0_) and low power instrument transformers (LPITs) (*f*_0_ ± 2%). Therefore, both inductive ITs and LPITs should be tested in a wider frequency range equal to *f*_0_ ± 15% (worst case of power frequency deviation variation).

#### 2.1.2. Supply Voltage and Current Deviation

The supply voltage (current) deviation is defined as the change of root mean square (RMS) value of the voltage (current) at a given time at the supply terminal, measured over a given interval. According to [16], the voltage and current amplitudes can vary between ±15% in a no-synchronous connection system. However, in the presence of very common PQ phenomena, such as dips, swells, interruptions, and transient overvoltage, the power grid amplitude can assume values in a wider range (from 1% to 200%). Standards on VTs and LPVTs require testing from 80% to 120% of their rated amplitude. Conversely, for the CTs and LPCTs, the tests prescribed by the standards are from 5% to 120% for devices intended for measuring applications and from 1% to 120% for protection applications. By combining the information above, it follows that the current ITs testing points are not sufficient to estimate their error contributions in the measurement of the amplitude during PQ phenomena, such as dips, swells, interruptions and overvoltage. For this reason, additional test points must be considered to cover the real voltage and current amplitude variation range. For VTs and LPVTs, a possible test range could be from 5% to 200%, whereas for CTs and LPCTs, a range from 1% to 200% would be appropriate.

#### 2.1.3. Harmonics and Interharmonics

A harmonic is a sinusoidal component with a frequency equal to an integer multiple of the fundamental frequency of the supply voltage. An interharmonic, rather, is a sinusoidal component with a frequency not equal to an integer multiple of the fundamental frequency. According to [3], considering the very short-term effects, the harmonic and interharmonic voltage can range from 0.2% to 8% in a frequency range from DC up to 9 kHz. The standard [28] on LPITs prescribes accuracy verification tests to evaluate their performance in harmonic measurements. This standard defines the harmonic order as testing points but does not provide information on harmonics amplitudes. Moreover, the standard [28] does not define the harmonic test waveform but it only suggests performing tests with a signal composed of a voltage/current at the rated frequency and amplitude plus a percentage of the rated primary input signal at each considered harmonic frequency. As the inductive ITs, the standards [11,12] do not require assessment of their accuracy at frequencies different from the rated one. At the same time, the main output from the literature review is that measurement of the harmonics and interharmonics can be significantly affected by an inductive IT because of its filtering behavior (mainly observed for VTs), as well as the non-linearity introduced by its ferromagnetic iron core [5].

For this reason, inductive VTs should be tested to evaluate their performances in the harmonics and interharmonics measurement. The test frequency range should be identified starting from the measurement/knowledge of the first resonance frequency (e.g., from a sinusoidal sweep test at low voltage). As for the CTs, a better frequency response up to 9 kHz is expected with respect to VTs, but non-linearities, depending on the iron core characteristics, can affect lower frequency harmonic responses. For both inductive ITs and LPITs, harmonics and interharmonics tests should be performed with realistic waveforms composed, for example, by a fundamental tone and one superimposed harmonic/interharmonic/tone or multi-tones waveform. In both cases, the fundamental amplitude should be set at the rated frequency and rated amplitude whereas the harmonic and interharmonic amplitudes should be chosen considering the limits set by the standards [16].

#### 2.1.4. Amplitude and Phase Modulations

An amplitude-modulated signal is a sinusoidal signal with an amplitude that is variable over time according to a specific function, for example, a sinusoidal waveform at frequency *f*_AM_. The case of a sinusoidal carrier at frequency *f*_0_ modulated with a sinusoidal waveform at frequency *f*_AM_ corresponds to the special case of a multitoned signal composed by the fundamental tone at *f*_0_ and two interharmonics at *f*_0_ ± *f*_AM_. The phase-modulated signal is a sinusoidal signal with a phase that varies in time according to a defined function, for example, a sine wave. These types of disturbances are introduced by the standard [29] and are intended for the evaluation of phasor measurement unit (PMU) performances under dynamic conditions. They are not considered in standards dealing with ITs. However, the literature [30] shows that inductive CTs can significantly affect the measurement of synchrophasor in the presence of amplitude and phase modulations. For this reason, both ITs and LPITs should be tested in the presence of modulations. The possible parameters of this type of test can be derived from those identified for PMU dynamic performance verification, which are:Amplitude modulation: Modulating frequency from 0.1 Hz to 5 Hz, with an amplitude equal to 10%.Phase modulation: Modulating frequency modulating from 0.1 Hz to 5 Hz, with modulating amplitude equal to 0.1 rad.

#### 2.1.5. Oscillatory Transients

An oscillatory transient is a short duration overvoltage, usually highly damped and with a duration of a few milliseconds or less. The detection and measurement of this phenomenon can be strongly influenced by the IT transient response and the amplitude dependence of its performance. Therefore, ITs should be tested in the presence of transient overvoltage. However, tests should be performed with maximum transient overvoltage at 200% and focusing on the phenomena characterized by a spectral content between 1 kHz and 9 kHz. The standard [15] describes oscillatory waves with frequencies as low as 300 Hz. Indeed, oscillatory transients observed in power systems, caused by capacitor energizing, restrike during capacitor de-energizing, or line or cable energizing, show typical frequencies of a few hundred Hz up to slightly above 1 kHz [31].

### 2.2. Literature and Standard Regarding ITs

Since ITs are the most used devices in the power network with the main purpose of adapting the level of amplitude voltage and current to input range of measurement devices at power frequency, the standards regarding ITs [11,12] are mainly focused on their characterization at 50 Hz or 60 Hz. At present, the accuracy of ITs in the measurement of PQ phenomena is not completely covered by international standards. The only document that deals with this topic is a technical report [13]. However, these documents do not include test methods, measurement setup, or measurement performance indices to evaluate the accuracy of ITs for PQ measurements. At present, there are few papers in the literature dealing with metrological performances of ITs used for PQ measurement. Some authors of this paper have already shown that the metrological performances of ITs in the presence of PQ phenomena can drastically change [30,32,33,34,35,36].

The main results of these studies can be summarized as follows:-A standard that covers ITs for PQ measurements in terms of performance indices and measurement setup does not exist.-The characterization of ITs described in international standards to characterize the ITs at power frequency is not suitable for the characterization of IT for PQ measurement.-The test waveform should be complex because the error of ITs can be drastically increased when more PQ phenomena are superimposed to fundamental waveform.-The simple and consolidated compensation techniques [33,34,35] present in the literature are ineffective when several PQ phenomena are superimposed to fundamental waveform.

The results of analysis of the literature and standards regarding PQ presented in this section show that the limits of international standards include the range of variation of PQ phenomena obtained in measurement campaigns. For this reason, the range of variation of considered PQ phenomena, shown in Table 1 and obtained from the literature and standard review, is the proposed range of variation for ITs testing in the sense of IT accuracy verification in PQ measurements.

## 3. Proposed Performance Indices for IT Characterization

This subsection introduces the proposed accuracy indices, namely performance indices (PIs), used to evaluate the ITs accuracy in the measurement of stationary, dynamic and transient PQ phenomena, referring specifically to the special case of voltage sensors. However, the performance indices proposed in this section can also be applied to current sensors, using current quantities instead of voltage quantities.

### 3.1. Steady-State Tests

For steady-state tests, two different categories of indices are defined.

Indices for the evaluation of the VT accuracy at specified harmonic or interharmonic frequency $\bar{f}$, i.e., the ratio error ε($\bar{f})$ and phase error $\Delta \varphi \left(\bar{f}\right)$: (1)$$\varepsilon \left(\bar{f}\right)=\frac{k_{r}\;U_{s}\left(\bar{f}\right)-U_{p}\left(\bar{f}\right)}{U_{p}\left(\bar{f}\right)}$$
(2)$$\Delta \varphi \left(\bar{f}\right)=\varphi_{s}\left(\bar{f}\right)-\varphi_{p}\left(\bar{f}\right)$$
where *k*_r_ = *U*_p,r_/*U*_s,r_ is the rated scale factor SF (*U*_p,r_ and *U*_s,r_ are the rated primary and secondary voltages); $U_{p}\left(\bar{f}\right)$ and $U_{s}\left(\bar{f}\right)$ are the RMS values of the primary and secondary voltage at the frequency $\bar{f}$; $\varphi_{p}\left(\bar{f}\right)$ and $\varphi_{s}\left(\bar{f}\right)$ are phase angles of the primary and secondary *h*-order harmonic voltage.

A synthetic index to quantify the VT performance over a specific frequency range, i.e., the TFrD error: (3)$$\varepsilon_{TFrD}=\frac{k_{r}TFrD_{s}-TFrD_{p}}{TFrD_{p}}$$
where $TFrD_{s}$ and $TFrD_{p}$ are defined as in the following Equations:(4)$$TFrD_{p}=\frac{\sqrt{\sum_{f=f_{1}}^{f_{N}}U_{p}^{2}\left(f\right)}}{U_{p}\left(f_{rated}\right)}$$
(5)$$TFrD_{s}=\frac{\sqrt{\sum_{f=f_{1}}^{f_{N}}U_{s}^{2}\left(f\right)}}{U_{s}\left(f_{rated}\right)}$$

The frequency amplitudes $U_{p}\left(\bar{f}\right)$ and $U_{s}\left(\bar{f}\right)$ and phases $\varphi_{p}\left(\bar{f}\right)$ and $\varphi_{s}\left(\bar{f}\right)$ are obtained by performing a discrete Fourier transform (DFT) over non-overlapped time frames equal to 10 cycles of the fundamental frequency, according to international standard [37].

### 3.2. Dynamic Tests

The performances in the presence of dynamic disturbances are evaluated using test waveforms indicated in the PMU standard [29]. For this reason, some of the proposed ITs performance indices are based on the indices used for the PMU characterization: the ratio error, the phase error, the total vector error (TVE), the frequency error (FE) and the rate of change of frequency (ROCOF) error (RFE), are expressed in the following Equations.
(6)$$\varepsilon =100\;\cdot \;\frac{k_{r}\left|V_{s}\right|-|V_{p}|}{\left|V_{p}\right|}$$
(7)$$\Delta \varphi =∡V_{s}-∡V_{p}$$
(8)$$TVE=\sqrt{\frac{{\left(Re\left(k_{r}V_{s}\right)-Re\left(V_{p}\right)\right)}^{2}+{\left(Im\left(k_{r}V_{s}\right)-Im\left(V_{p}\right)\right)}^{2}}{Re{\left(V_{p}\right)}^{2}+Im{\left(V_{p}\right)}^{2}}}$$
(9)$$FE=f_{0,s}-f_{0,p}$$
(10)$$RFE=\frac{df_{0,s}}{dt}-\frac{df_{0,m}}{dt}=ROCOF_{s}-ROCOF_{p}$$
where *k*_r_ = *V*_p,r_/*V*_s,r_ is the rated scale factor SF (*V*_p,r_ and *V*_s,r_ are the rated primary and secondary voltages); ***V_p_*** and ***V_s_*** are the fundamental voltage phasors at VT primary and secondary side, respectively. The quantities *f*_0,p_, *f*_0,s_ are the fundamental frequencies measured at VT primary and secondary side, respectively.

The fundamental primary and secondary phasors (***V_p_*** and ***V_s_***) are estimated via DFT on an observation interval of four cycles of the fundamental frequency (i.e., 0.08 s). A reporting rate of 50 (60) Hz or 100 (120) Hz, for a 50 (60) Hz power system is used.

### 3.3. Oscillatory Transients

The proposed test waveform for oscillatory transients of short duration can be modeled as an exponentially decaying sinusoid with initial onset:(11)$$v_{OT}\left(t\right)=\sqrt{2}\;U_{\mathrm{OT}}\;\sin\left(2\pi f_{OT}t+\varphi_{OT}\right)\;\cdot \;e^{-t/\tau }\;$$

The Equation (11) is characterized by their initial peak magnitude value $U_{\mathrm{OT}}$, oscillation frequency $f_{OT}$, initial phase $\varphi_{OT}$ of the damped sinewave, and decay time τ. These values can only be accurately defined and determined after filtering the fundamental component of waveform The proposed corresponding performance indices are:
The change in first peak magnitude value *U*_pk_ = $\sqrt{2}U_{\mathrm{OT}}$
(12)$$\varepsilon_{\mathrm{Upk}}=100\;\cdot \;\left(\frac{U_{\mathrm{pk},s}-U_{\mathrm{pk},p}}{U_{\mathrm{pk},p}}-1\right)\;\;$$Oscillation frequency $f_{OT}$ of the damped sine wave.Phase displacement (or time shift) of the damped sine wave.Decay time τ of the oscillation.


These parameters can be analyzed in the time domain after filtering the 50 Hz component by fitting the damped sinusoid waveform. Alternatively, one can determine them as follows:The first peak magnitude value $\sqrt{2}U_{\mathrm{OT}}$ can be estimated as the maximum of the observed measurement values.The oscillation frequency $f_{OT}$ can be obtained from successive zero-crossings.The phase displacement can be calculated as the difference in zero-crossings following the initial peak values.The decay time τ can be obtained by fitting an exponential decay $e^{-t/\tau }$ to the successive peak values.

The errors made by these approximations are negligible because, for all parameters, the difference between, or the ratio of, the values obtained for the reference device and the IT under test is considered. This way, the error is made twice and will be canceled out.

### 3.4. Summary

To briefly recap, this subsection provides, through Table 2, a summary of the test types, the quantities to measure and the related indexes.

## 4. Measurement Setups for VTs and CTs Characterization

This section describes the measurement setups for the characterization of the ITs under test. The rated characteristics of the investigated ITs are shown in Table 3. The first is an inductive VT, resin insulated with primary and secondary winding and a ferromagnetic core. The second is a LPVT based on capacitive technology and without a ferromagnetic core. Both the VTs are for MV grids. In this paper, for the sake of brevity and clarity, only results related to the characterization of voltage transformers are shown. 

In fact, the focus of the paper is the proposals of new test procedures, test waveforms and indexes to evaluate the error contribution of ITs to PQ measurements. Therefore, the experimental results are presented for the validation of the proposal. However, the proposed procedure can still be applied to the characterization of current transformers.

### 4.1. VTs Characterization

The measurement setup used for inductive MV VT characterization is shown in Figure 1a. The signal generation is obtained by the NI PXI 5422 (AWG board, with 16-bit, variable output gain, ±12 V output range, 200 MHz maximum sampling rate, 256 MB onboard memory). Another NI PXI 5412 is used to generate a 12.8 MHz clock, which is used as time base clock for the comparator. The 10 MHz PXI clock is used as a reference clock for both AWG boards. The voltage waveform generated by the AWG is amplified by a Trek high-voltage power amplifier (30 kV, 20 mA, peak values) with wide bandwidth (from DC to 2.5 kHz at full voltage and 30 kHz at reduced voltages), high slew rate (<550 V/µs) and low noise. Applied voltage reference values are obtained by means of a 30 kV wideband reference divider, designed, built, and characterized at INRIM. The acquisition system is obtained through NI cDAQ chassis with four different acquisition modules: NI 9225 (±425 V, 24-bit, 50 kHz), NI 9227 (±14 A, 24-bit, 50 kHz), NI 9239 (±425 V, 24-bit, 50 kHz), NI 9238 (±500 mV, 24-bit, 50 kHz).

The measurement setup used for LPVT characterization is shown in Figure 1b. The reference voltage signal to be applied to the LPVT under test is provided by NI PXI 5422. Acquisition of the primary and secondary waveforms of the VT under test has been performed through the data acquisition board PXIe-6124 (±10 V, 16 bit, maximum sampling rate of 4 MHz). Waveforms have been sampled with a sampling rate of 100 kHz obtained through oversampling in order to reduce the impact of noise. The output of the AWG is connected to a high-voltage power amplifier (NF HVA4321, up to 10 kV, from 0 Hz up to 30 kHz) feeding the VT under test. Primary voltages are scaled by Ohm-Labs KV-10A High Voltage Divider with rated voltage of 10 kV, ratio of 1000 V/V and accuracy below 0.1%.

### 4.2. CTs Characterization

The setups of a reference system for current sensor calibrations at power frequency up to 5 kA and for the wideband frequencies up to 9 kHz are illustrated in Figure 2 (more details in [18]). The calibration system is mainly made up of a current generation system [19] (marked as a red block), a set of reference current-to-voltage (C-to-V) transformers (marked as green blocks in the path N), the device under test (DUT, marked as yellow blocks in the path X) and a high precision two-channel measuring system (MS, marked as a grey block). The DUT shown as a general C-to-V transformer in Figure 2 represents any type of current sensors under test such as an inductive current transformer under test with a reference measuring resistor connected to the secondary, a Rogowski coil with or without active integrator or a high-current shunt. The rated current-to-voltage ratio of the DUT *F*_n,X_ is determined by the ratio of the rated output voltage *U*_n,X_ over the rated input current *I*_P,n_ (*F*_n,X_ = *U*_n,X_/*I*_P,n_) The laboratory capability of the current generation system at power frequency ranges from 5 A to 5 kA. The wideband generation capability is evaluated with frequencies ranging from 50 Hz to 5 kHz with 100 A. Dual-tone or multi-tone waveforms up to 1 kA as well as amplitude-modulated waveforms are generated. The restriction at frequencies greater than 500 Hz for 1 kA is due to the inherent stray inductance of the high current generating transformer. The reference C-to-V transformer with rated primary currents ranging from 8.3 A to 1500 A for wideband calibrations up to 12 kHz is composed of a symmetrical CT and associated precise measuring resistor (more details in [19]). The accuracies of the reference CTs are within 10 µA/A and 1 μrad at power frequency. The frequency response of the CTs up to 12 kHz were below 0.1% and 2 mrad with the expanded uncertainties below 0.01% and 0.3 mrad (k = 2). To convert the secondary currents of the diverse CTs into a 1 V output voltage, a resistor box was built, containing six self-developed precision resistors *R*_m_ from 1 Ω to 20 Ω. The calibrated results of each resistor are published in [18].

The precise two-channel measuring system used for the CT calibrations at power frequency has a Type B uncertainty of about 10^−6^ (more details in [20]). To connect the secondary side of CTs (N and X) to the MS, a reference current-to-voltage converter [21] (rated input current 100 mA to 5 A, uncertainty of below 1 µA/A and 1 µrad) is used. The precise two-channel measuring system used for the wideband calibrations has basic uncertainties of 3 µV/V or μrad at 50 Hz under various proposed PQ phenomena in Section 5.1 and uncertainties of below 20 µV/V or μrad at frequencies up to 20 kHz. The detailed errors and uncertainties of the measuring system by 50 Hz with input voltage ranging from 0.1 V to 3 V under various PQ phenomena are listed in Table 4.

#### Compensated Current Comparator for Inductive CT Calibration in a Wider Frequency Range

A wideband characterization of a compensated current comparator (CCC), built at CMI, and normally employed as a standard in power frequency calibration has been performed to assess its usability as an alternative reference sensor for CT characterization. As a result, the CCC allows CT characterization up to 9 kHz with uncertainties up to 400 µA/A and 0.30 µrad for the ratio and phase, respectively. 

Its applicability to the calibration 100 A/5 A inductive CT has been demonstrated and the generation and measurement setup used is provided in Figure 3.

### 4.3. Combined ITs Characterization

The measurement setup used for the characterization of MV combined ITs and sensors is shown in Figure 4. The signal generation is obtained by six synchronized items of programmable power sources (Kikusui PCR series with the power output ranges of 500 W, 1 kW, 2 kW and 6 kW). One master unit externally connected to the other five slave units are synchronized and controlled either via on-board or by a computer program. High power voltage outputs (up to 300 V of fundamental) of these power sources are directly applied to the primary windings of three units of high voltage power amplifiers and to the primary windings of three units of high transconductance power amplifiers to have high voltages up to 36 kV and currents up to 2 kA, respectively.

A CT bridge and a VT bridge are to be operated for accuracy measurements while a wideband bridge and an analyzer with wideband reference current and voltage sensors are set for PQ measurements and analysis of influence factors such as proximity effects. This three-phase setup allows generation of multiple external magnetic and electrical fields at the same time and their alignment in three dimensions.

All required performance tests were performed, covering the signal generation and conditioning, bandwidth and power limits of each source, synchronization criteria, amplifier types and their behaviors in long-term operations such as stability and losses, isolation in cabling up to 36 kV, verification of bridges for sinusoidal and non-sinusoidal waveforms.

## 5. Proposed Tests and Experimental Results

### 5.1. Proposed Test Waveforms and Test Points

#### 5.1.1. Amplitude and Frequency Deviations

For this type of test, the proposed test waveform is a pure sinewave at amplitudes and frequency chosen according to the values given in Table 5:

#### 5.1.2. Harmonics and Interharmonics

For these types of tests, two test waveforms are proposed, that are FT1 and FTN.

The FT1 test signal waveform consists of two voltage components: a fundamental component at power frequency and one superimposed tone at frequency $f_{T}$. The FT1 signal is mathematically described by Equation:(13)$$v_{\mathrm{FT}1}\left(t\right)=\surd 2U_{n}\sin\left(2\pi f_{0}t\right)+\surd 2U_{T}\sin\left(2\pi f_{T}t+\varphi_{T}\right)$$
where, *U*_n_, *f*_0,_ *U*_T_, $\varphi_{T}$ are the RMS value of the rated primary voltage, the power frequency, the RMS value of component at the frequency $f_{T}$, and the initial phase angle of the component at frequency $f_{T}$, respectively.

The FT1 parameters chosen for the presented experimental tests are detailed in Table 6, referring to the specific cases of harmonics and interharmonics.

The FTN test signal is an extension of the FT1 one. In this case, the voltage waveform consists of a multi-tone signal composed by a fundamental component at power frequency plus *N* superimposed tones. The mathematical description is provided in Equation:(14)$$v_{\mathrm{FTN}}\left(t\right)=\surd 2U_{n}\sin\left(2\pi f_{0}t\right)+\surd 2\overset{N}{\underset{k=1}{\sum }}U_{T,k}\sin\left(2\pi f_{T,k}t+\varphi_{T,k}\right)$$
where, *U*_n_, *f*_0,_ *U*_T,k_, $\varphi_{T,k}$ are the RMS value of the rated primary voltage, the power frequency, the RMS value of the harmonic or interharmonic frequency $f_{T,k}$, and the initial phase angle of the component at frequency $f_{T,k}$, respectively. The FTN parameters chosen for the experimental tests are detailed in the Table 7, referring to the specific cases of harmonics and interharmonics:

#### 5.1.3. Amplitude-Modulated Signal

The proposed test waveform used for the IT performance evaluation under amplitude-modulated signal is described in the Equation (15). It is composed by a fundamental tone at power frequency and a sinusoidal amplitude modulating signal:(15)$$v_{AM}\left(t\right)=\sqrt{2}U_{n}\left(1+k_{\mathrm{AM}}\cos\left(2\pi f_{\mathrm{AM}}t\right)\right)\;\cdot \;\cos\left(2\pi f_{0}t\right)$$
where *U*_n_, *f*_0,_ *k*_AM_, and *f*_AM_ are the RMS value of the rated primary voltage, the frequency of the fundamental component, the value of modulation factor and the modulation frequency, respectively.

The amplitude modulation parameters generated for the experimental tests are summarized in Table 8.

#### 5.1.4. Phase-Modulated Signal

The proposed test waveform is described in the following Equation:(16)$$v_{PM}\left(t\right)=\sqrt{2}U_{n}\cos\left[\left(2\pi f_{0}t\right)+k_{\mathrm{PM}}\cos\left(2\pi f_{PM}t-\pi \right)\right]$$
where *k*_PM_ and *f*_PM_ are the phase modulation factor and the phase modulation frequency, respectively.

The phase modulation test parameters chosen for the experimental tests are given in Table 9:

#### 5.1.5. Oscillatory Transient

The proposed test waveform for the VT performance evaluation under oscillatory transient conditions is described in the Equation (17). It is composed of a fundamental tone plus a superimposed damped sinusoidal signal:(17)$$v_{OT}\left(t\right)=\sqrt{2}U_{n}\sin\left(2\pi f_{0}t\right)+\sqrt{2}U_{\mathrm{OT}}\sin\left(2\pi f_{OT}t+\varphi_{OT}\right)\cdot e^{-t/\tau },$$
where $U_{\mathrm{OT}}$, $f_{OT}$ and $\varphi_{OT}$ are the initial amplitude, the frequency and the phase of the damped sine wave, respectively, and 1/τ is the decay constant.

Note that for CTs it is possible to apply the waveform described in Equation (17). However, often due to geomagnetically induced currents there are unipolar currents that flow in the transmission system and appear as DCs that exponentially decrease [38]. For this reason, in the case of CTs the waveform (17) can be substituted by Equation (18).
(18)$$i_{OT}\left(t\right)=I_{DC}e^{-\frac{t}{\tau_{DC}}}+\sqrt{2}I_{n}\sin\left(2\pi f_{0}t\right)+\sqrt{2}I_{\mathrm{OT}}\sin\left(2\pi f_{OT}t+\varphi_{OT}\right)\cdot e^{-t/\tau }$$
where $I_{DC}$ and $\tau_{DC}$ are the amplitude and and the decay constant of the DC component, respectively.

The generated oscillatory transients are described by the parameters provided in Table 10.

### 5.2. VTs Characterization: Experimental Results

This subsection provides the experimental results related to the inductive VT and LPVT characterization under the test waveforms and test points indicated in Section 5.1.

The VT performances have been assessed by performing measurement with the generation measurement setup shown in Figure 1a, whereas the LPVT with the setup in Figure 1b.

The quantities at the VTs primary and secondary sides are acquired with a sampling frequency equal to 50 kHz and 100 kHz, for the setup in Figure 1a,b. The time window chosen is 1 s and ten repetitions are executed for each test.

#### 5.2.1. Amplitude and Frequency Variations for the Inductive VT

The inductive VT ratio and phase errors under FT1 conditions at different fundamental amplitudes and frequencies are reported in Figure 5a,b.

As can be observed, the inductive VT has quite a linear behavior in terms of both amplitude and frequency. Considering the curves with the black-square marker (A = 100%), the ratio error transitions from −0.01% at 42.5 Hz to 0.08% at 57.5 Hz. A similar result is found for the phase error that is equal to 0.25 mrad at 42.5 Hz and it transitions to −0.20 mrad at 57.5 Hz. 

With regard to the amplitude dependence, considering the 50 Hz frequency case, it can be observed that at 5% the ratio and phase errors are equal to −0.04% and 0.20 mrad and their absolute values decrease to 0% and 0.01 mrad at 120%.

#### 5.2.2. Harmonics and Interharmonics for the Inductive VT

The inductive VT performance in the measurement of harmonics and interharmonics are reported in Table 11 according to PIs (1) and (2) introduced in Section 3. Table 11 provides the results in the measurement under FT1 conditions whereas Figure 6a,b shows the ratio and phase errors versus frequency under FTN conditions.

As can be observed, the VT introduces quite high errors at the third harmonics (−0.84% for the ratio and 1.27 mrad for the phase). These errors are mainly due to the non-linear behavior of the VT iron core. In fact, if the frequency considered in the analysis is not the third harmonics (150 Hz) but a slightly different frequency (149 Hz), the errors significantly decrease (0.03% for the ratio and −1.07 mrad for the phase). This is due to the spurious component introduced at the third harmonic frequency due to the non-linear behavior of the ferromagnetic core of the inductive VT under test. 

For higher frequencies, a predominant filtering behavior due to the stray capacitance is observed. Considering the PI $\varepsilon_{TFrD}$ (3), the VT introduces an error equal to −1.07% when it is supplied with the harmonic Test Point 4 and it is equal to −0.56% under the interharmonic Test Point 4.

#### 5.2.3. Amplitude and Phase Modulations for the Inductive VT

The inductive VT does not show significant variations of ratio and phase error in the presence of amplitude and phase modulation. Under these test conditions, additional PIs are evaluated according to Section 3: the TVE, RFE and FE errors. These errors reach the maximum values of 0.57%, 150 µHz and 3 mHz/s, respectively.

#### 5.2.4. Oscillatory Transient for the Inductive VT

The errors introduced by the inductive VT in the measurement of the oscillatory transients are evaluated in the time domain after removing the fundamental component, as shown in Figure 7.

The resulting VT PIs are summarized in Table 12. As can be observed, the errors associated with the measurement of the peak value and with the decay time increase with the increase in the frequency of the oscillations *f*_OT_. As for the time shift, only in two cases (*f*_OT =_ 500 Hz and 5000 Hz) are the zero-crossings following the initial peak values found at different times.

#### 5.2.5. Amplitude and Frequency Variations for the LPVT

The ratio and phase errors of the LPVT at various fundamental amplitudes and frequencies are shown in Figure 8a,b.

As it can be seen, the errors of the LPVT ranges from −0.3% at 57.5 Hz and 5% of the fundamental amplitude up to 0.45% at 42.5 Hz and 120% of fundamental amplitude, and from −4.3 mrad at 42.5 Hz and 120% of fundamental amplitude up to 3.2 mrad at 57.5 Hz and 5% of fundamental amplitude, respectively, for ratio and phase errors.

The LPVT shows quite a linear behavior in terms of amplitude and frequency variation. Unlike the VT behavior shown in the previous section, the LPVT has a higher phase error, even if the value is under the limits of its accuracy class (0.5%, 9 mrad).

#### 5.2.6. Harmonics and Interharmonics for the LPVT

The LPVT performance in the measurement of harmonics and interharmonics are reported in Table 12 according to PIs (1) and (2) introduced in Section 3. In particular, Table 13 provides the results in the measurement under FT1 conditions, whereas Figure 9a,b shows the frequency ratio and phase errors under FTN conditions.

As can be observed in Figure 9, the LPVT introduces higher errors than the inductive VT when the frequency increases. In fact, the ratio and phase errors reach up to 5.56% and 1.2 rad, respectively, at 2500 Hz. Moreover, the same behavior can also be observed for interharmonic components. These behaviors are probably due to the construction technology of the tested LPVT, essentially based on capacitive elements [36].

#### 5.2.7. Amplitude and Phase Modulations for the LPVT

The LPVT performance in the measurement of TVE, FE, and RFE are reported in Table 14 according to PIs (8–10) introduced in Section 3. In particular, Table 14 provides the maximum values of PIs when amplitude and phase modulation occur with 5 Hz and 2 Hz modulation frequencies, respectively. As it can be seen, the LPVT introduces additional errors to the indices typical of the PMUs. As it can be seen from Figure 10a,b, the RFE index can strongly vary in the presence of an LPVT with regard to the amplitude and phase modulation, respectively.

Furthermore, the RFE index is strongly influenced by the phase modulation with a modulation frequency of 2 Hz, as shown in Figure 10b. In fact, the RFE index ranges from −40 Hz/s up to 40 Hz/s; on the other hand, in the case of phase modulation with a modulation frequency of 5Hz, the effect is reduced by more than 50%. In the case of amplitude modulation, the difference between the modulation frequency of 5 Hz and 2 Hz is negligible. However, the RFE in the case of amplitude modulation ranges from −13 Hz/s up to 14 Hz/s. This represents a particularly critical situation: in fact, according to [29], the RFE limit for a M class PMU in the modulation test is equal to 14 Hz/s. Therefore, using a measurement chain for synchrophasor measurements composed by a capacitive LPVT, such as the LPVT tested here, and a PMU should be considered. The complete chain, in some situations, exceeds the RFE limits of the standard [29], even if the RFE of the PMU alone is under the limit of the standard. This fact, in particular, suggests that great attention should be paid to the choice of instrument transformers for power quality measurements.

#### 5.2.8. Oscillatory Transient for LPVT

The PIs associated with the LPVT under test in the measurement of oscillatory transients are summarized in Table 15. For the LPVT, the errors increase with the increase in the frequency of the oscillations, according to the frequency response shown in Figure 9a. The worst case is observed for *f*_OT_ = 5000 Hz where the LPVT introduce and error in the measurement of the decay time τ close to −5.5%.

## 6. Conclusions

This paper has presented a proposal for IT accuracy verification tests used for PQ measurements. Starting from the deep review of enforced standards and the scientific literature, a set of PQ phenomena relevant for IT testing along with their range of variation are chosen. Possible performance indices and test waveforms, with particular attention to amplitude and frequency deviation, harmonics and interharmonics, amplitude and phase modulation and oscillatory transient, are defined. The proposed approach is validated by applying it to the accuracy verification of two commercial MV VTs, specifically an inductive VT and a capacitive LPVT.

Experimental results show that both the inductive VT and the LPVT under test remain in their accuracy class in the presence of amplitude and frequency variations. With regard to the error introduced in the measurement of harmonics and interharmonics, the inductive VT errors are generally lower than those introduced by the LPVT. In particular, up to 2500 Hz, the VT ratio and phase frequency responses are within −3% and −8 mrad, whereas for the LPVT the observed values are −6% and −1.2 mrad. Focusing on the measurement of harmonics, the inductive VT shows a non-linear behavior in the measurement of low order odd harmonics, whereas the LPVT is more linear. With regard to the amplitude and phase modulation tests, the PIs associated with the inductive VT are very low, which means that the tested VT does not significantly influence the measurement of the fundamental synchrophasor in the presence of modulations. Conversely, the error introduced by the LPVT under test in the presence of modulation are significant, especially the RFE index. Finally, both the tested ITs show errors in the measurement of transient oscillations peak value and decay time that increase with the increase in the oscillation frequency.

## Figures and Tables

**Figure 1 sensors-22-05847-f001:**
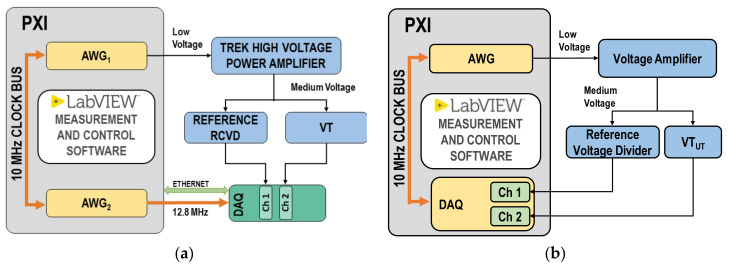
Block diagram of the generation and measurement setup: (**a**) for inductive MV VT; (**b**) for LPVT.

**Figure 2 sensors-22-05847-f002:**
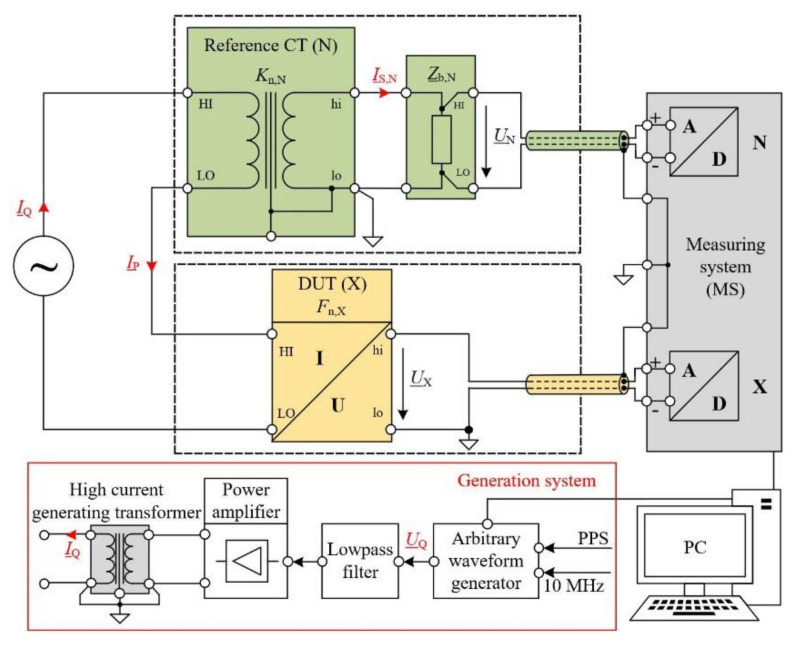
Block diagram of the generation and measurement setup for current sensor under test.

**Figure 3 sensors-22-05847-f003:**
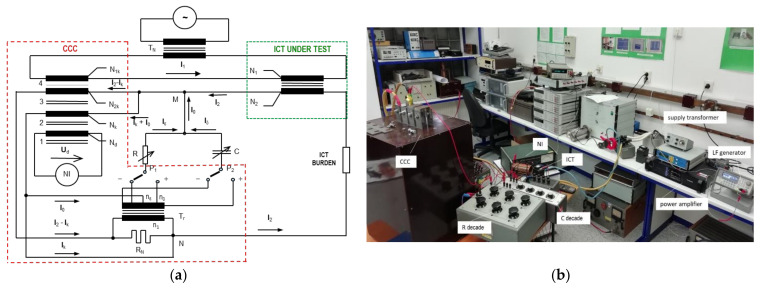
Block diagram (**a**) and picture (**b**) of the generation and measurement setup for inductive CT characterization with CMI CCC.

**Figure 4 sensors-22-05847-f004:**
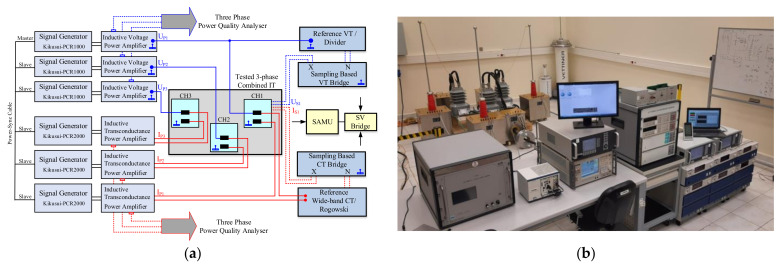
Block diagram (**a**) and picture (**b**) of the generation and measurement setup for MV combined ITs.

**Figure 5 sensors-22-05847-f005:**
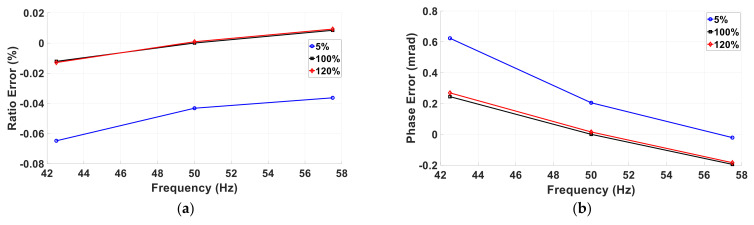
Ratio (**a**) and phase (**b**) errors of the VT under test at different fundamental amplitudes and frequencies.

**Figure 6 sensors-22-05847-f006:**
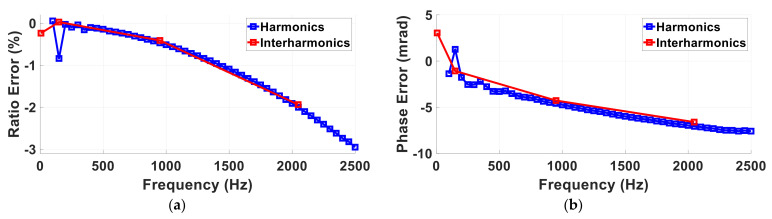
Ratio (**a**) and phase (**b**) errors of VT under test in FTN conditions with harmonic components (circle markers) and interharmonic components (square markers).

**Figure 7 sensors-22-05847-f007:**
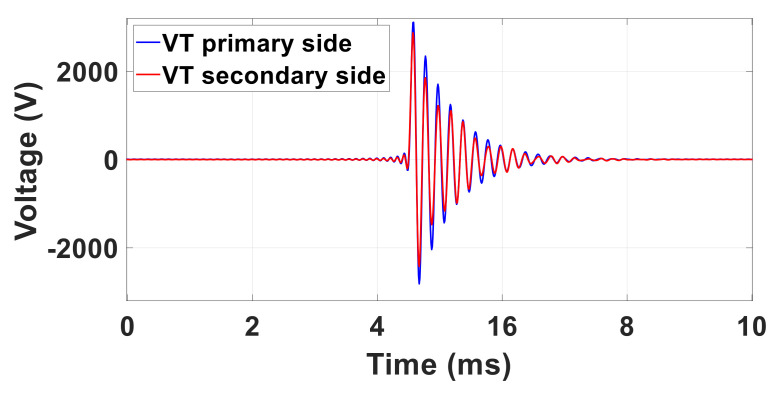
Oscillatory transient characterized by *U*_OT_ = 22% of *U*_n_, *f*_OT_ = 5 kHz and τ = 600 µs, φ_OT_ = 0 rad measured at VT primary side (blue curve) and secondary side (red curve).

**Figure 8 sensors-22-05847-f008:**
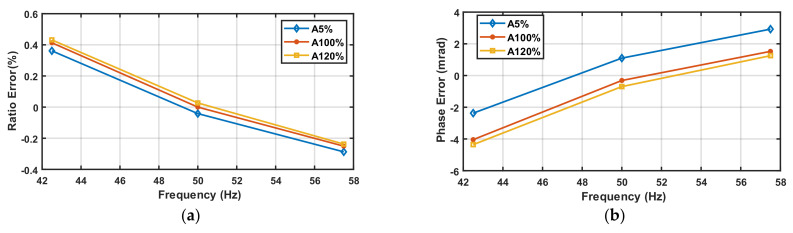
Ratio (**a**) and phase (**b**) errors of LPVT under test at different fundamental amplitudes and frequencies.

**Figure 9 sensors-22-05847-f009:**
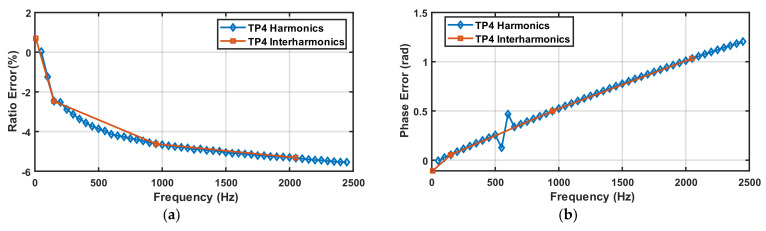
Ratio (**a**) and phase (**b**) errors of LPVT under test in FTN conditions with harmonic components (rhombus markers) and interharmonic components (square markers).

**Figure 10 sensors-22-05847-f010:**
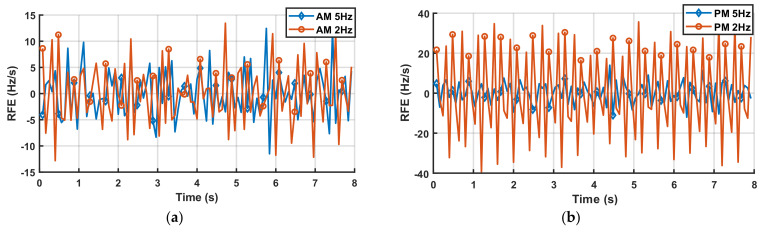
RFE of LPVT under test in amplitude (**a**) and phase (**b**) modulations with modulation frequency of 5Hz (rhombus markers) and 2Hz (circle markers).

**Table 1 sensors-22-05847-t001:** Maximum limit of PQ phenomena from the literature and standards.

PQ Phenomenon	Limits
Frequency deviation	±15% of rated frequency
Supply voltage and current deviation	From 5% up to 200% of amplitude rated voltage From 1% up to 200% of amplitude rated current
Harmonic voltage	10% from 2nd up to 15th—5% from 16th up to 50th 2% from 51th up to 9 kHz
Interharmonic voltage	3% from DC up to 20 Hz—5% from 20 Hz up to 100 Hz 1% from 100 Hz up to 9 kHz
Amplitude and phase modulation	Frequency modulating from 0.1 Hz up to 5 Hz—K_x_ = 0.1% Frequency modulating from 0.1 Hz up to 5 Hz—K_a_ = 0.1 rad
Oscillatory Transient	Up to 5 kHz, up to 22% of rated amplitude

**Table 2 sensors-22-05847-t002:** Proposed PIs for IT characterization: a summary.

Test Category	Test Type	Quantity to Measure	Performance Index
Steady State	Amplitude and Frequency Variation	Amplitude	Ration error $\varepsilon \left(\bar{f}\right)$
		Phase	Phase error $\Delta \varphi \left(\bar{f}\right)$
	Harmonics and Interarmonics	Amplitude	Ratio error $\varepsilon \left(\bar{f}\right)$
		Phase	Phase error $\Delta \varphi \left(\bar{f}\right)$
		Total Distortion	Total frequency error $\varepsilon_{TFrD}$
Dynamic	Amplitude modulation Phase modulation Frequency Ramp	Amplitude	Ratio error ε
		Phase	Phase error Δφ
		Combination of amplitude and phase	Total Vector Error *TVE*
		Frequency	Frequency Error *FE*
			Rate of change of Frequency Error *RFE*
Transient	Oscillatory Transient	Peak magnitude	Error peak magnitude $\varepsilon_{\mathrm{Upk}}$
		Time shift	Time shift error $\Delta t_{\mathrm{zero}-\mathrm{crossing}}$
		Decay time	Devay time error $\varepsilon_{\tau }$

**Table 3 sensors-22-05847-t003:** Rated characteristics of the investigated instrument transformers.

Name	Primary (kV)	Secondary (V)	Rated Burden (VA)	Accuracy Class	Rated Insulation Level (kV)
Inductive VT	20/√3	100√3	50	0.5	12
LPVT	7	7	25	0.5	24

**Table 4 sensors-22-05847-t004:** The errors and uncertainties of the two-channel measuring system at 50 Hz with input voltage ranging from 0.1 V to 3 V under various PQ phenomena.

	Single Sinusoidal	Harmonics/Interharmonics	Amplitude/ Phase Modulation	Transient
*ε*(*f*_0_) in µV/V	0	3	1	1
*δ*(*f*_0_) in µrad	0	1	1	1
*U*(*ε*_0_) in µV/V	3	3	3	3
*U*(*δ*_0_) in µrad	2	2	1	1

*ε*(*f*_0_) refers to the ratio errors at 50 Hz. Δ(*f*_0_) refers to the phase errors at 50 Hz. *U*(*ε*_0_) refers to the expanded uncertainties for ratio errors. *U*(*δ*_0_) refers to the expanded uncertainties for phase errors.

**Table 5 sensors-22-05847-t005:** Proposed test point for amplitude and frequency deviations.

	Test Point 1			Test Point 2			Test Point 3		
	1.A	1.B	1.C	2.A	2.B	2.C	3.A	3.B	3.C
Frequency (Hz)	42.5	50	57.5	42.5	50	57.5	42.5	50	57.5
Amplitude (% of rated)	5			100			120		

**Table 6 sensors-22-05847-t006:** Proposed test point for amplitude and frequency deviations.

**Harmonics**	**Test Point 1**	**Test Point 2**	**Test Point 3**
Amplitude (% of fundamental)	5	10	1
Harmonic Order	2nd	3rd	50th
**Interharmonics**	**Test Point 1**	**Test Point 2**	**Test Point 3**
Amplitude (% of fundamental)	5	10	1
Frequency (Hz)	75	375	2475

* Fundamental component at rated amplitude and frequency.

**Table 7 sensors-22-05847-t007:** Proposed test point for harmonics and interharmonics.

	Test Point 4
**Harmonics**	Harmonics at 1% of the fundamental from the 2nd to the 50th order
**Interharmonics**	1% of the fundamental at 7 Hz, 149 Hz, 951 Hz, 2048 Hz

* Fundamental component at rated amplitude and frequency.

**Table 8 sensors-22-05847-t008:** Proposed test point for amplitude modulation.

Amplitude Modulation	Test Point 1	Test Point 2
*k*_AM_ (% of fundamental)	10	10
*f*_AM_ (Hz)	2	5

* Fundamental component at rated amplitude and frequency.

**Table 9 sensors-22-05847-t009:** Proposed test point for phase modulation.

Phase Modulation	Test Point 1	Test Point 2
*k*_PM_ (rad)	0.1	0.1
*f*_PM_ (Hz)	2	5

* Fundamental component at rated amplitude and frequency.

**Table 10 sensors-22-05847-t010:** Proposed test point for oscillatory transient.

Oscillatory Transient	Test Point 1	Test Point 2	Test Point 3	Test Point 4
$U_{\mathrm{OT}}$ (% of fundamental)	22	22	22	22
*f*_OT_ (Hz)	500	1000	2000	5000
τ (µs)	600	600	600	600

* Fundamental component at rated amplitude and frequency.

**Table 11 sensors-22-05847-t011:** Ratio and phase errors of VT in harmonics and interharmonic measurements.

	Harmonics		Interharmonics	
	$\varepsilon \left(hf_{0}\right)$ (%)	$\Delta \varphi \left(hf_{0}\right)$ (mrad)	$\varepsilon \left(\bar{f}\right)$ (%)	$\Delta \varphi \left(\bar{f}\right)$ (mrad)
Test Point 1	0.045	−0.93	0.0158	−0.49
Test Point 2	−0.062	−1.02	−0.042	−2.41
Test Point 3	−2.93	−7.35	−2.86	−7.23

**Table 12 sensors-22-05847-t012:** VT PIs results under oscillatory transient phenomena.

	$\varepsilon_{U_{pk}}$ (%)	$\Delta t_{\mathrm{zero}-\mathrm{crossing}}$ (ms)	$\varepsilon_{\tau }$ (%)
Test Point 1 *f*_OT_ −500 Hz	−0.04	0.02	−0.36
Test Point 2 *f*_OT_ −1000 Hz	−0.06	0	−0.49
Test Point 3 *f*_OT_ −2000 Hz	−3.59	0	−1.01
Test Point 4 *f*_OT_ −5000 Hz	−7.61	0.02	−4.52

**Table 13 sensors-22-05847-t013:** Ratio and phase errors of LPVT in harmonics and interharmonic measurements.

	Harmonics		Interharmonics	
	$\varepsilon \left(hf_{0}\right)$ (%)	$\Delta \varphi \left(hf_{0}\right)$ (rad)	$\Delta \varepsilon \left(\bar{f}\right)$ (%)	$\Delta \varphi \left(\bar{f}\right)$ (rad)
Test Point 1	−1.42	0.026	−0.85	0.01
Test Point 2	−2.20	0.056	−3.55	0.18
Test Point 3	−5.56	1.2	−5.57	1.22

**Table 14 sensors-22-05847-t014:** Maximum values of TVE, FE and RFE of LPVT.

	Amplitude Modulation			Phase Modulation		
	TVE (%)	FE (mHz)	RFE (Hz/s)	TVE (%)	FE (mHz)	RFE (Hz/s)
Test Point 1	1.85	3.2	12.44	1.95	3.9	13.97
Test Point 2	1.88	4.1	13.43	2.05	10.4	39.27

**Table 15 sensors-22-05847-t015:** LPVT PIs results under oscillatory transient phenomena.

	$\varepsilon_{U_{pk}}$ (%)	$\Delta t_{\mathrm{zero}-\mathrm{crossing}}$ (ms)	$\varepsilon_{\tau }$ (%)
Test Point 1 *f*_OT_ −500 Hz	−3.86	0.03	−2.35
Test Point 2 *f*_OT_ −1000 Hz	−5.10	0.04	−3.42
Test Point 3 *f*_OT_ −2000 Hz	−5.74	0.04	−4.75
Test Point 4 *f*_OT_ −5000 Hz	−8.72	0.03	−5.57

## Data Availability

Not applicable.

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
