# Peer review of "How Instrument Transformers Influence Power Quality Measurements: A Proposal of Accuracy Verification Tests"

_sensors, 2022, doi:10.3390/s22155847_

Round 1

Reviewer 1 Report

The authors have presented a very interesting and well-conceived work. It relates both theoretical content and experimental measurements, presenting all the research carried out in a didactic way. I would like to congratulate the researchers for the manuscript presented. Only a few minor comments are indicated:

Line 22: Do not use this type of abbreviations in the abstract: "f.i.".

The number of keywords is excessive and should be reduced as indicated in the Template.

Describe the meaning of the acronym "EMPIR".

In section 2, line 112, 120, etc. Use subsections 2.1.1. or 2.1.2. according to the Template. Use this recommendation in all sections.

Section 2 is very interesting and enriching. Surely, the incorporation of some explanatory schematic/circuit could be included.

Line 248, correct reference [2828].

The results are clear and concise, with a very elaborate experimental program. I would recommend placing the discussion of the Figures once they have appeared in the text.

The last section of Conclusions is very brief. The authors have done an excellent job with a large number of significant results and implications derived from their experimentation. This section should be expanded.

Author Response

All the revisions to the paper are in red color.

All the responses to the reviewers are in red color.

Please see the attachment for the responses to the reviewers.

Reviewer 2 Report

This article is very interesting and deals with important issues related to the analysis of energy quality. The structure of the article is correct. There is everything that is needed in a good article: a review of the literature, description of the problem, description of its solution and the results of tests carried out on a real research object.

Also in editorial terms, the article is of a high standard, although I did find a few bugs which are listed below.

Most of them concern references to literature. Examples below.

Line 173: "[2929]" - Entry 2929 is missing in the list of references. Maybe the comma after 29 is missing, but referring to the same item twice is also a mistake.

Line 204: "[29,3129-36]" - Entry 3129 is missing from the literature list. Here it is probably enough to delete entry 31 as it is in the range 29-36.

Line 248: "[2828]" - comment as above

Line 349: "[2019]" - should it be [19, 20]?

And one more minor typo:

Line 288: "...procedure can still be applied to che..." - Shouldn't it be "the"?

Author Response

(The authors gave the same response as above.)

Reviewer 3 Report

A set of the PQ-related phenomena regarding the voltage and current instrumentation transformers are covered – in terms of standards, prepositions of specific measuring points and verified on voltage transformers.

1.       Abbreviations LPIT and LPVT are not known till Conclusion.

2.       How do the inductive VT and LPVT differ? Please specify their main parameters.

3.       Reference quoting typo in lines 173, 204, 248,  349…

4.       Table 1: reconsider to change: “Transient oscillator” into “Oscillatory Transients”

5.       Missing punctuation in line 266

6.       Typo in line 288

7.       Reconsider deleting Table 1 in “published in [22] in TABLE I” – line 342 in order to reduce ambiguity.

8.       Unit crad in lines 338 and 339.  Use mrad or urad instead.

9.       Change “conventional” to “ inductive” in line 342

10.   “C to V converter” in line 348: A better description would be: “two-channel A/D converter with input C to V conversion”. Otherwise, the original description is similar to the one in line 334.

11.   Footnote under table 3: s(ε0), s(δ0) Typo? U instead of s?

12.   Figure 4a: please recheck the symbol for CH_n (separate current and voltage branches)

13.   “the RMS value of the frequency ?T” in line 402: RMS value?? A similar mistake in line 420.

14.   Abbreviation PI is not declared in text – line 468.

15.   Line 474: The text in the paragraph is unclear to which measuring situation (FT1/FTN) belongs.

16.   Line 477: The reason for significant improvement in the case of 149Hz remains somehow undefined. Namely, does it depend also on the applied measuring concept, i.e. DFT and windowing effects?

17.   Typo “meanly” line 475.

18.   Section 3 instead Section 4 in line 488

19.   Delete “a higher phase ratio” in line 511

20.   Typo “resting”? in line 560

The authors claim that the applied characterization and proposed test waveforms described in the case of voltage transformers apply correspondingly current instrumentation transformers. Namely, the waveform (17) does not take into account a DC transient phenomenon, which is far more often in current transformer measurements. Please, provide an explanation or possible expansion of the chapter.

Author Response

(The authors gave the same response as above.)

Reviewer 4 Report

Title: A Proposal of Accuracy Verification Tests of Instrument Transformers Used for Power Quality Measurements

Comments:

Refine the title. It is not attractive

Abstract is general. Please make it focused and improve it to show your contribution and main findings

Reduce the keywords. Maximum is ten

Define the research gap in introduction

Stat specific contribution at the end of introduction

Why only results related to the characterization of voltage transformers are presented. Justify

Why the sampling frequency is to 100 kHz. Justify

Conclusion is general. Please make it focused and improve it to show your contribution and main findings

Author Response

(The authors gave the same response as above.)

Round 2

Reviewer 3 Report

Well done!

Reviewer 4 Report

The paper can be accepted